# Coupled small molecules target RNA interference and JAK/STAT signaling to reduce Zika virus infection in *Aedes aegypti*

**Chasity E. Trammell**[1], **Gabriela Ramirez**[2], **Irma Sanchez-Vargas**[2], **Laura A. St Clair**[2], **Oshani C. Ratnayake**[2], **Shirley Luckhart**[3,4], **Rushika Perera**[2]*, **Alan G. Goodman**[1,5]*

1 School of Molecular Biosciences, College of Veterinary Medicine, Washington State University, Pullman, Washington, United States of America, 2 Center for Vector-borne Infectious Diseases, Department of Microbiology, Immunology and Pathology, Colorado State University, Fort Collins, Colorado, United States of America, 3 Department of Entomology, Plant Pathology, and Nematology, College of Agricultural and Life Sciences, University of Idaho, Moscow, Idaho, United States of America, 4 Department of Biological Sciences, College of Science, University of Idaho, Moscow, Idaho, United States of America, 5 Paul G. Allen School for Global Health, College of Veterinary Medicine, Washington State University, Pullman, Washington, United States of America

* rushika.perera@colostate.edu (RP); alan.goodman@wsu.edu (AGG)

**Data Availability Statement:** All relevant data are within the manuscript and its Supporting Information files.

## Abstract

The recent global Zika epidemics have revealed the significant threat that mosquito-borne viruses pose. There are currently no effective vaccines or prophylactics to prevent Zika virus (ZIKV) infection. Limiting exposure to infected mosquitoes is the best way to reduce disease incidence. Recent studies have focused on targeting mosquito reproduction and immune responses to reduce transmission. Previous work has evaluated the effect of insulin signaling on antiviral JAK/STAT and RNAi in vector mosquitoes. Specifically, insulin-fed mosquitoes resulted in reduced virus replication in an RNAi-independent, ERK-mediated JAK/STAT-dependent mechanism. In this work, we demonstrate that targeting insulin signaling through the repurposing of small molecule drugs results in the activation of both RNAi and JAK/STAT antiviral pathways. ZIKV-infected *Aedes aegypti* were fed blood containing demethylasterriquinone B1 (DMAQ-B1), a potent insulin mimetic, in combination with AKT inhibitor VIII. Activation of this coordinated response additively reduced ZIKV levels in *Aedes aegypti*. This effect included a quantitatively greater reduction in salivary gland ZIKV levels up to 11 d post-bloodmeal ingestion, relative to single pathway activation. Together, our study indicates the potential for field delivery of these small molecules to substantially reduce virus transmission from mosquito to human. As infections like Zika virus are becoming more burdensome and prevalent, understanding how to control this family of viruses in the insect vector is an important issue in public health.

## Author summary

Arboviruses pose a significant threat to humans and are an increasing concern as a result of climate change and expanding vector-competent populations. The recent Zika

**Funding:** This research was supported by the WSU College of Veterinary Medicine Stanley L. Adler research fund to A.G.G, NIH / National Institute of General Medical Sciences (NIGMS)-funded pre-doctoral fellowship (T32 GM008336) and a Poncin Fellowship to C.E.T., NIH/NIAID grant R01AI151166 to I.S.V. and R.P., and University of Idaho and UI College of Agricultural and Life Sciences startup funds to S.L. The funders had no role in study design, data collection and analysis, decision to publish, or preparation of the manuscript.

**Competing interests:** The authors have declared that no competing interests exist.

outbreaks demonstrate that mosquito-borne illnesses caused by viral infection remain a prominent and evolving threat that must be actively addressed. As there are currently no post-exposure therapeutics available for Zika virus infection, reducing transmission and, in turn, the likelihood of infection would provide sizeable benefit for human populations most at risk. Here, we show that readily available small molecules can be repurposed to effectively reduce viral replication and likelihood of transmission for a clinically relevant strain of Zika in vector competent mosquitoes. Furthermore, we show how two insulin-mediated canonical antiviral pathways are simultaneously activated in our drug treatment regimen to reduce virus levels in mosquito saliva, through which virus is transmitted to humans. Together, we demonstrate the viability of targeting insulin signaling as a means of reducing the rate of mosquito infection and decreased transmission of Zika virus.

## Introduction

Mosquito-borne viruses pose a significant global health threat, and this threat is increased by dynamic ecological and human factors. Global warming and urbanization have permitted mosquitoes and arboviruses to spread into regions that previously lacked mosquito infection and mosquito-to-human virus transmission [1,2]. This occurred during the 2015–17 Zika virus (ZIKV) Western hemisphere epidemic that originated in South America and spread into North America, resulting in 538,451 suspected cases, 223,477 confirmed cases, and 3,720 congenital syndrome cases [3,4]. Subsequent outbreaks have followed that establishes ZIKV as an active pathogen of concern that requires intervention [5]. Current efforts have focused on strategies to reduce virus transmission to and from the mosquito vector, including the use of insecticides and biological/genetic manipulation of primary vector species. The introduction of the bacterial symbiont *Wolbachia* to reduce flavivirus infection in the major arbovirus vector species *Aedes aegypti* [6–8] and the release of genetically modified individuals to reduce transmission-competent progeny of this species [9] have been included among the latter strategies. There is evidence to suggest that *Wolbachia*, while effective in reducing ZIKV and dengue virus (DENV) infection in targeted species may inadvertently enhance replication of West Nile virus (WNV) [10]. It is also not known how effective or advantageous genetically modified mosquito populations are compared to wild type populations or to other various viruses [11,12]. Because of these challenges, additional strategies to reduce vector transmission of these important viral pathogens are critically needed.

As an alternative strategy, it may be possible to reduce or block arborvirus transmission through mosquito-targeted delivery of bioactive small molecules at attractive sugar bait stations, a modification of the successful delivery of toxic baits for mosquito control [13]. To this end, it is necessary to identify druggable mosquito antiviral effectors and their upstream regulatory factors. The insulin/insulin-like growth factor signaling (IIS) cascade regulates RNA interference (RNAi) and JAK/STAT antiviral immunity against West Nile virus (WNV), dengue virus (DENV), and ZIKV [14,15]. In *Drosophila melanogaster*, the IIS-dependent transcription factor forkhead box O (FOXO) induces expression of RNAi transcripts *AGO2* and *Dicer2* [16]. We demonstrated that ingestion of exogenous insulin reduced expression of these RNAi components in WNV-infected *Culex quinquefasciatus* [14] and that manipulation of IIS-dependent extracellular-signal regulated kinases (ERK) activation reduced WNV infection in this mosquito host [14]. Further, insulin treatment suppressed the activation of RNAi, while activating ERK-dependent JAK/STAT induction of unpaired (upd) ligands to control WNV replication *in vitro* and *in vivo* [14]. Previous studies established that both JAK/STAT and

RNAi antiviral pathways are independently involved in arthropod antiviral immunity to ZIKV [17–20]. To date, however, no mechanism(s) have been established whereby both antiviral pathways are induced simultaneously in response to arthropod infection.

In this study, we repurposed small molecules that target IIS pathway proteins to induce simultaneous activation of RNAi and JAK/STAT signaling in *Ae. aegypti*. Specifically, we used the potent insulin mimetic demethylasterriquinone B1 (DMAQ-B1), an activating ligand of the insulin receptor (InR) [21] and Protein kinase B (AKT) inhibitor VIII, which reduces AKT phosphorylation [22] (**Fig 1A**). Small molecule treatment induced activation of JAK/STAT via ERK and blocked inhibition of RNAi via the AKT/FOXO signaling axis. Combined treatment with DMAQ-B1 and AKT inhibitor VIII significantly lowered ZIKV titers in *Ae. aegypti* cells and adult female mosquitoes relative to single treatment and vehicle control. Combined treatment also additively reduced salivary gland virus titers, a surrogate measure of reduced transmission efficacy [23,24]. Accordingly, we argue that activation of both antiviral pathways resulted in enhanced defenses that lowered viral titers to non-detectable levels. This work demonstrates the feasibility of strategically targeting mosquito immunity via IIS as a means of reducing a clinically relevant strain of ZIKV infection and transmission at the vector level.

## Results

### DMAQ-B1 and AKT inhibitor VIII activated *Aedes aegypti* insulin and antiviral signaling pathways

Since the JAK/STAT and RNAi antiviral pathways are linked to IIS, we sought to test the activity of small molecules against phosphorylation of key IIS protein targets and activity of these antiviral pathways. Given that phosphorylation of AKT and ERK correlate with activation of IIS and JAK/STAT signaling [14,25], respectively, and that FOXO phosphorylation is consistent with suppression of RNAi [16,26,27], we used these readouts to evaluate the efficacy of DMAQ-B1 and AKT inhibitor VIII control of RNAi and JAK/STAT.

Protein lysates from *Ae. aegypti* Aag2 cells treated with 1% DMSO vehicle, 1 μM DMAQ-B1, 10 μM AKT inhibitor VIII, or these combined drug treatment for 24 hours were analyzed by western blot for phosphorylation of AKT, FOXO, and ERK (**Fig 1B**). Drug concentrations were based on prior toxicity analysis (**S1 Fig**). DMAQ-B1 treatment was associated with the highest levels of AKT and FOXO phosphorylation; this phosphorylation was significantly reduced when combined with AKT inhibitor VIII (**Fig 1C and 1D**). Single drug and combined drug treatments were associated with increased ERK phosphorylation relative to vehicle control (**Fig 1E**). We validated these findings by immunofluorescence microscopy of 24 h treated cells probed for phospho-FOXO (P-FOXO) and P-ERK. Consistent with western blot analyses, we observed increased P-FOXO only in the DMAQ-B1-treated cells and P-ERK in both individual and combined-treated cells (**Fig 1F**). Further, P-FOXO localization was primarily cytosolic (**Fig 1G**) in DMAQ-B1 treated cells and P-ERK localization was primarily nuclear (**Fig 1H**) in AKT inhibitor and combined treated cells, confirming that the transcription factors involved in RNAi and JAK/STAT induction are both nuclear and transcriptionally active under the expected treatment conditions. We also observed increased transcript expression of *AGO2* and *virus-induced RNA-1* (*vir*-1) which are indicative of RNAi and JAK/STAT activation, respectively, in cells treated with the combined drugs (**Fig 1I and 1J**). Collectively, these data indicated that DMAQ-B1 and AKT inhibitor VIII treatment alter IIS phosphorylation in *Ae. aegypti* cells in a pattern consistent with the activation of FOXO- and ERK-dependent antiviral signaling independent of viral infection.

Based on these effects on RNAi and JAK/STAT signaling in the absence of virus, we sought to determine the effects of single and combined drugs on ZIKV replication in *Ae. aegypti* cells.

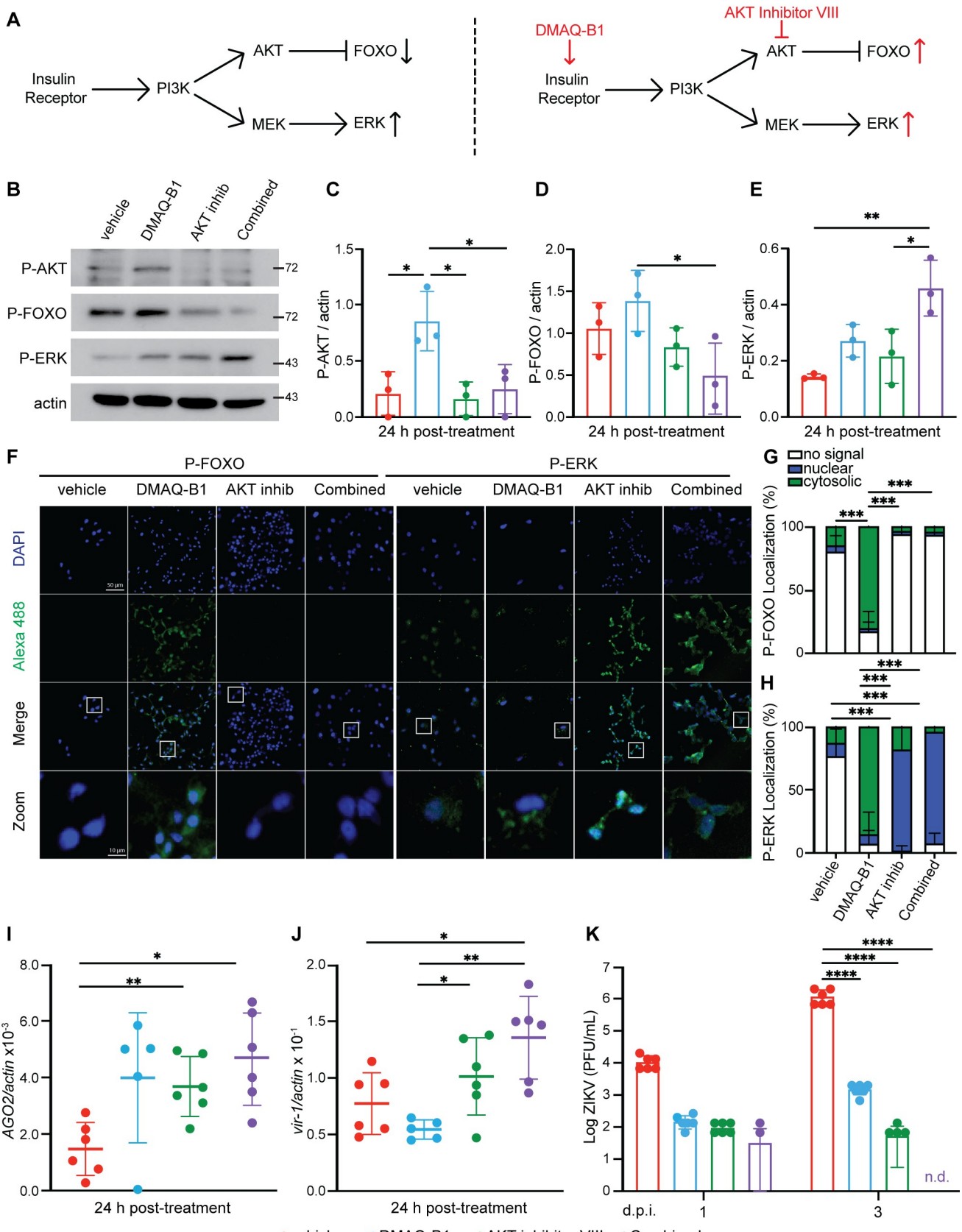

**Fig 1. DMAQ-B1 and AKT inhibitor VIII activated RNAi and JAK/STAT *in vitro*.** (A) Insulin/insulin-like growth factor I signaling (IIS) pathway schematic and proposed effect of DMAQ-B1 and AKT inhibitor VIII on downstream transcription factor activity. (B-E) Aag2 cells were treated with

vehicle (DMSO), 1 μM DMAQ-B1, 10 μM AKT inhibitor VIII, or combined drugs for 24 h. Phosphorylation of AKT, FOXO, and ERK were measured by western blot and phosphorylation was quantified for (B) P-AKT, (C) P-FOXO, and (D) P-ERK by densitometry and normalized to actin (*p < 0.05, One-way ANOVA with Tukey's test correction for multiple comparisons). (F) P-FOXO and P-ERK abundance and localization were visualized in DAPI-stained Aag2 cells by immunofluorescence microscopy. (G-H) Protein localization of (F) P-FOXO and (G) P-ERK was quantified in microscopy images to evaluate whether fluorescent-tagged proteins was cytosolic or nuclear within individual cells by manually counting cells in images (***p<0.001, Two-way ANOVA with Tukey's correction). (I-J) Induction of RNAi and JAK/STAT signaling was evaluated as transcript levels of (I) *AGO2* and (J) *vir-1* by qRT-PCR (*p < 0.05; **p<0.01, Unpaired t test with Welch's correction for multiple comparisons). (K) Aag2 cells that received vehicle, individual, or combined drug treatment for 24 h were infected with ZIKV (MOI 0.01) and supernatant was collected at 1 and 3 d p.i. Supernatant virus was titered by standard plaque assay (****p<0.0001, Two-way ANOVA with Tukey's correction). Closed circles represent individual replicates. Horizontal bars represent mean and error bars represent SD. Results are representative of triplicate independent experiments.

Aag2 cells were primed with individual and combined drugs for 24 h prior to infection with the clinically isolated PRVABC59 strain of ZIKV (**Fig 1H**). We observed significant reductions in ZIKV titer in cells treated with individual and combined drugs by 3 days post-infection (d p.i.). Most notably, ZIKV titers were undetectable by 3 d p.i. in cells treated with the combined drugs (**Fig 1H**). Patel and Hardy (2012) showed that AKT inhibitor VIII was antiviral in Sindbis virus (SINV)-infected *Aedes albopictus* C6/36 cells [28], but the dysfunctional RNAi response in these cells [29] precluded the confirmation of mechanism in its entirety. Accordingly, we concluded that DMAQ-B1 and AKT inhibitor VIII treatments induced an antiviral response that was increased to the point of non-detectable ZIKV titers when these treatments were combined.

## DMAQ-B1- and AKT inhibitor VIII-supplemented sucrose water induces activation of RNAi and JAK/STAT signaling in *Aedes aegypti*

To model established sugar-bait strategies implemented in the field [13] and translatability of our *in vitro* findings, we next evaluated the effect of DMAQ-B1 and AKT inhibitor VIII treatment against the IIS-dependent antiviral response in adult female *Ae. aegypti*. Aged-matched 6-9-day old female mosquitoes were continuously exposed to 10% sucrose water supplemented with either 1% DMSO vehicle control, individual, or combined 10 μM DMAQ-B1 and 10 μM AKT inhibitor VIII. Survival was measured over a 14-day treatment period with no difference between treatment conditions (**S2 Fig**). Whole mosquitoes were collected at 3 (**Fig 2A–2D**), 7 (**Fig 2E–2H**), and 11 day (**Fig 2I–2L**) for analysis of RNAi and JAK/STAT gene induction by qRT-PCR. *AGO2* and *p400* were examined as markers of RNAi (**Fig 2A, 2B, 2E, 2F, 2I and 2J**) [30,31], while *Vago2* and *vir-1* were examined as downstream effectors of JAK/STAT (**Fig 2C, 2D, 2G, 2H, 2K and 2L**) [32,33]. At day 3 we observed higher expression of RNAi-related genes in combined drug-treated mosquitoes compared to our vehicle control (**Fig 2A and 2B**) but no difference in JAK/STAT-related genes (**Fig 2C and 2D**). At day 7, however, we saw that RNAi and JAK/STAT gene induction were significantly higher in our combined drug-treated mosquitoes compared to our vehicle control (**Fig 2E–2H**). By day 11 we observed a loss in immune gene induction on the combined drug-treated mosquitoes (**Fig 2I–2L**). This leads us to infer that the effectiveness and activity of DMAQ-B1 and AKT inhibitor VIII, at least in combination, induces RNAi and JAK/STAT genes is optimal up to 7 days of continuous treatment but gene induction is reduced by 11 days, possibly due to negative feedback that inhibits the induction of these genes during continuous feeding. Because of this limited toxicity, transcript profile, and range of efficacy, we conclude that DMAQ-B1 and AKT inhibitor VIII induces RNAi and JAK/STAT signaling in a similar manner as we observed *in vitro*.

## Bloodmeal treatment of DMAQ-B1 and AKT inhibitor VIII in ZIKV-infected *Aedes aegypti* induced simultaneous activation of RNAi and JAK/STAT signaling

Based on IIS-dependent antiviral activity of DMAQ-B1 and AKT inhibitor VIII *in vitro* and *in vivo*, we sought to evaluate whether similar drug effects could be detected in *Ae. aegypti* adult

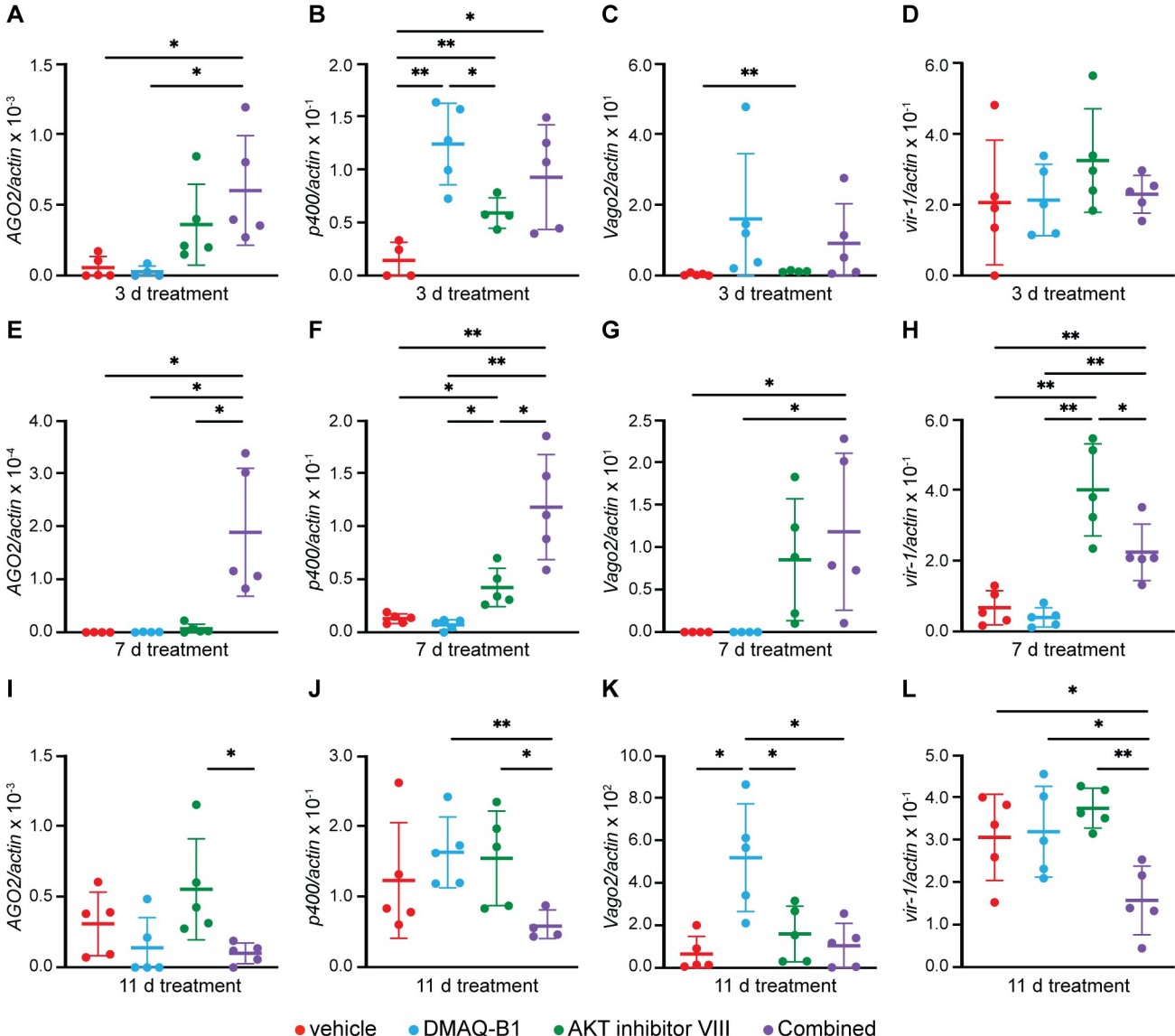

**Fig 2. Continuous drug treatment via sucrose water results in induction of RNAi and JAK/STAT signaling in *Aedes aegypti* mosquitoes.** Induction of RNAi and JAK/STAT gene transcripts at (A-D) 3 d, (E-H) 7 d, and (I-L) 11 d treatment was measured in whole mosquitoes receiving continuous treatment of vehicle, 10 μM DMAQ-B1, 10 μM AKT inhibitor VIII, or combined drugs via sucrose water. Transcripts were measured for (A, E, I) *AGO2*, (B, F, J) *p400*, (C, G, K) *Vago2*, and (D, H, L) *vir-1* by qRT-PCR (*p < 0.05; **p<0.01, Unpaired t test with Welch's correction for multiple comparisons). Closed circles represent individual replicates. Outliers were identified using a ROUT test (Q = 5%) and removed. Horizontal bars represent mean and error bars represent SD. Results represent duplicate independent experiments.

females during ZIKV infection. Aged-matched 6–9-day old female mosquitoes were fed a ZIKV-containing bloodmeal including vehicle, individual, or combined 10 μM DMAQ-B1 and 10 μM AKT inhibitor VIII. Drug concentrations were selected based on mortality studies to measure drug lethality to mosquitoes over a dose range (**S3 Fig**). Mosquitoes were collected at 3, 7, and 11 d p.i., timepoints that corresponded with complete digestion of the blood meal, progression of viremia into distal tissues, and virus infection of the salivary glands [34–36]. Expression levels of RNAi and JAK/STAT signaling gene products were quantified by qRT-PCR at 7 d p.i. (**Fig 3A–3D**) and 11 d p.i. (**Fig 3E–3H**). *AGO2* and *p400* were examined as markers of RNAi (**Fig 3A**, **3B**, **3E and 3F**) [30,31], while *Vago2* and *vir-1* were examined as

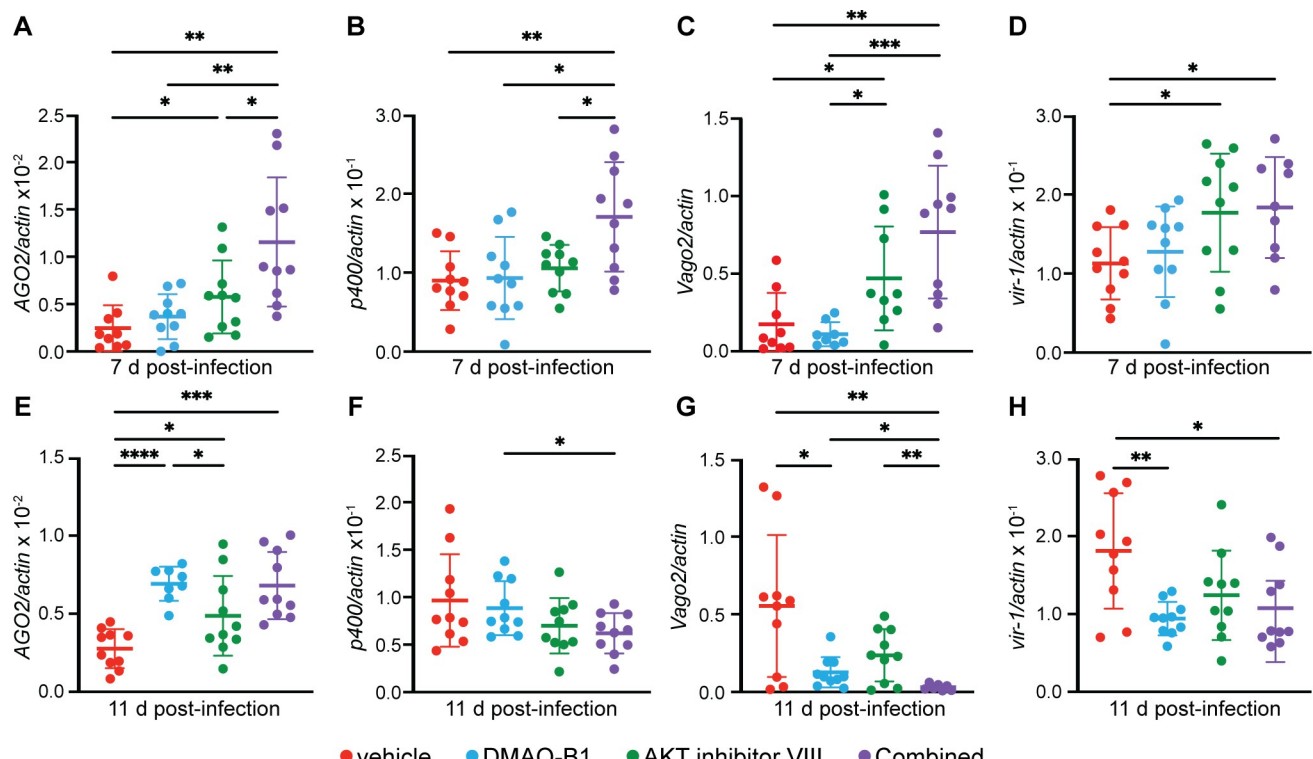

**Fig 3. Combined drug treatment induced activation of RNAi and JAK/STAT in *Aedes aegypti* at 7 d p.i. that was reduced by 11 d p.i.** Induction of RNAi and JAK/STAT gene transcripts at (A-D) 7 d p.i. and (E-H) 11 d p.i. was measured in whole mosquitoes infected with ZIKV and treated with vehicle, 10 μM DMAQ-B1, 10 μM AKT inhibitor VIII, or combined drugs. Transcripts were measured for (A, E) *AGO2*, (B, F) *p400*, (C, G) *Vago2*, and (D, H) *vir-1* by qRT-PCR (*p < 0.05; **p<0.01; ***p<0.001; **** p<0.0001, Unpaired t test with Welch's correction for multiple comparisons). Closed circles represent individual replicates. Outliers were identified using a ROUT test (Q = 5%) and removed. Horizontal bars represent mean and error bars represent SD. Results represent duplicate independent experiments.

downstream effectors of JAK/STAT (**Fig 3C, 3D, 3G and 3H**) [32,33]. Similar to previously discussed findings in Aag2 cells and non-infected mosquitoes, we observed that the combination drug treatment resulted in higher expression of RNAi and JAK/STAT signaling gene products at 7d p.i. (**Fig 3A–3D**). Interestingly, at 11 d p.i., only transcript levels for *AGO2* remained significantly higher for individual drug- and combination drug-treated mosquitoes (**Fig 3E–3H**). This gene expression profile is similar to mosquitoes that were continuously treated with individual or dual-drug supplemented sucrose water. Based on high transcript expression at 3 d p.i. in mosquitoes treated with the drug combination (**S4 Fig**), we demonstrate loss of gene induction between 7 and 11 d p.i. suggesting that drug treatment may have a limited efficacy by 11 d post-bloodmeal regardless if DMAQ-B1 and AKT inhibitor VIII is introduced in a single blood-feed or a multiday exposure. Expression levels of other related RNAi genes including *Dicer2* and *Ppo8* [30,37] and JAK/STAT *dome* [38] were enhanced in combined drug-treated mosquitoes 7 d p.i. (**S5 Fig**) which affirms the effect that DMAQ-B1 and AKT inhibitor VIII have for RNAi and JAK/STAT signaling in *Ae. aegypti* mosquitoes.

## DMAQ-B1 and AKT inhibitor treatment reduced infection prevalence and ZIKV titer in *Aedes aegypti*

We next sought to evaluate the effects of individual and combined drug treatments on infection prevalence and ZIKV titers in adult mosquitoes. Mosquitoes were fed a ZIKV-containing

bloodmeal treated with vehicle, DMAQ-B1, AKT inhibitor VIII, or combined drug treatment as described. We collected mosquitoes at 3, 7, and 11 d p.i. and analyzed individual midguts, pairs of salivary glands, and carcasses for ZIKV titers (n = 30). There were no differences in virus infection prevalence or viral titers at 3 d p.i. (**S6 Fig**). At 7 d p.i. infection prevalence was significantly reduced midgut and salivary gland tissues but not carcasses from combined drug-treated mosquitoes (**Fig 4A–4C**). The titer of ZIKV-positive mosquitoes at 7 d p.i., however, was only reduced in the midguts of combined drug-treated mosquitoes with an approximate 30-fold reduction. There was no difference in carcass and salivary gland titers (**Fig 4D–4F**). By 11 d p.i., both infection prevalence (**Fig 4G–4I**), and viral titer (**Fig 4J, 4K and 4L**) were significantly reduced across tissues in mosquitoes treated with individual and combined drug treatment relative to vehicle control. Titers in ZIKV-positive mosquitoes were reduced by ~42-fold in carcass, ~15-fold in midgut, and ~24-fold in salivary glands (**Fig 4J–4L**). Notably, infection prevalence and salivary gland viral titers were reduced in combined drug-treated mosquitoes at 11 d p.i., a time consistent with virus transmission during feeding [39,40]. Mosquitoes that received the combined drug treatment were not only less likely to be ZIKV-positive, but salivary gland viral load was also substantially reduced in positive mosquitoes. These observations suggested that combined drug treatment and coordinated activation of RNAi and JAK/STAT provides antiviral immunity against ZIKV that effectively reduced infection prevalence and viral load compared to vehicle controls. These effects of combined drug treatment would be predicted, therefore, to reduce mosquito transmission of ZIKV.

## Inhibition of RNAi and JAK/STAT signaling resulted in loss of drug-mediated antiviral protection

To confirm that DMAQ-B1 and AKT inhibitor VIII mediated-antiviral protection is via RNAi- and JAK/STAT-dependent responses, we transfected Aag2 cells with siRNA constructs to knockdown expression of *AGO2* and *vir-1*. Cells were transfected with siRNAs that targeted either gene individually (siAGO2, siVir-1) or stacked gene expression (siAGO2 + siVir-1) [41]. We measured *AGO2* and *vir-1* expression at 48, 72, and 120 h in accordance with siRNA removal, when cells were infected, and when supernatant was collected, respectively (**Fig 5A**). We observed significantly reduced gene expression at 48 h and 72 h post transfection for both individual and stacked siRNA treatments compared to cells that were treated with non-targeting siRNAs as a control [42]. Individual siAGO2 treated cells exhibited a 68% and 41% reductions in *AGO2* expression at 48 h and 72 h, respectively, whereas stacked siRNA treated cells exhibited 84% and 99% reduction at the same timepoints (**Fig 5B**). We observed that *AGO2* expression returned to control levels in our siAGO2 and siAGO2 + siVir-1 transfected cells by 120 h. Expression levels of *vir-1* in siVir-1- and stacked transfected cells were reduced at all timepoints measured (**Fig 5C**). In siVir-1-transfected cells, *vir-1* expression was reduced by 69% at 48 h, 88% at 72 h, and 74% at 120 h. In stacked-transfected cells, *vir-1* expression was reduced by 82% at 48 h, 77% at 72 h, and 63% at 120 h. Finally, we treated cells at 48 h post siRNA transfection with vehicle, individual, or combined drug treatments for 24 h prior to ZIKV infection. Viral titers were measured in supernatants collected at 2 d p.i. to determine if drug-mediated antiviral protection was impacted in the absence of antiviral RNAi, JAK/STAT, or both. We observed that both individual and combined *AGO2* and *vir-1* knockdowns resulted in significant losses of drug-mediated antiviral protection relative to drug-treated controls (**Fig 5D**). Interestingly, while we observed a loss in antiviral protection when comparing all data among gene knockdown groups (**S2 Table**, **Sheet1**), we observed a significant increase in viral titers only among the vehicle control groups when comparing siControl to siVir-1 or combined knockdown (**S2 Table**, **Sheet2**). One explanation for why we did not observe

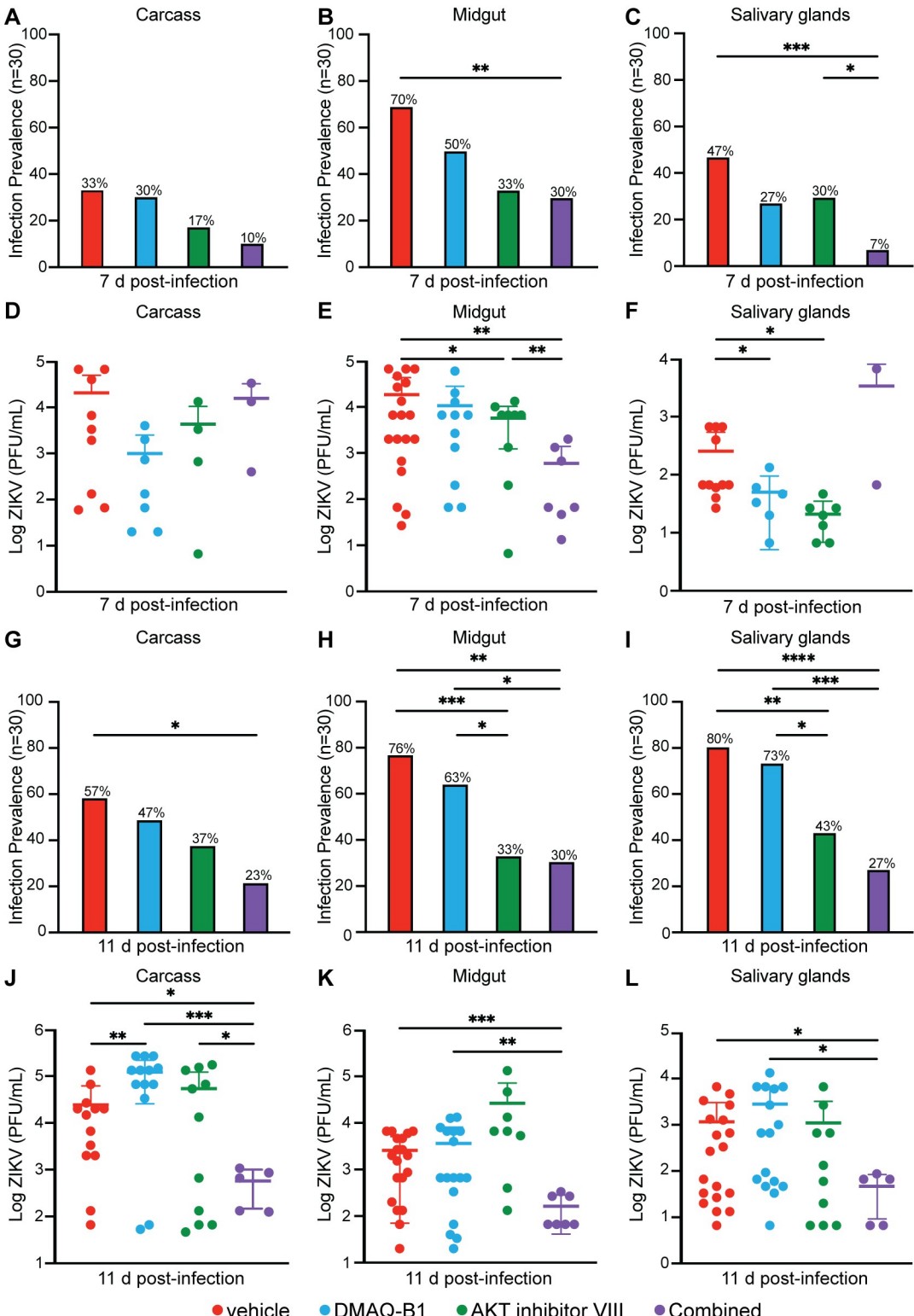

**Fig 4. Individual and combined drug treatments reduced infection prevalence and ZIKV titers in *Aedes aegypti*.**
Individual mosquito midguts, pairs of salivary glands, and carcasses (n = 30) were titered for ZIKV by standard plaque assay at
(A-F) 7d p.i. and (G-L) 11 d p.i. (A-C, G-I) Infection prevalence was calculated as the ratio of ZIKV-positive samples to the
total sample size (*p < 0.05; **p<0.01; ***p<0.001, Two-tailed Fisher's exact test). (D-F, J-L) Viral titers were determined in
ZIKV-positive mosquitoes (*p<0.05; **p<0.01; ***p<0.001; **** p<0.0001, Unpaired t test with Welch's correction for

multiple comparisons). Closed circles represent individual replicates. Outliers were identified using a ROUT test (Q = 5%) and removed. Horizontal bars represent mean and error bars represent SD. Results represent duplicate independent experiments.

differences between the siControl and siAGO2 group during vehicle treatment may be that while AGO2/RNAi is inhibited at the time of infection, its activity is restored by the time of sample collection (**Fig 5B**). Another possibility is that while IIS-dependent RNAi and JAK/STAT signaling are sufficient to significantly reduce ZIKV titers, other pathways may also contribute to this biology. For example, despite the reduction in ZIKV to undetectable levels via IIS-dependent antiviral immunity, Toll signaling [20] and autophagy [43] could contribute to control of ZIKV replication. Collectively, our data suggest that repurposing small molecule drugs to target mosquito IIS can induce antiviral responses that significantly reduce ZIKV infection prevalence and transmission potential in *Ae. aegypti* through activation of RNAi and JAK/STAT signaling as inhibition of either or both pathways result in a loss of drug-mediated antiviral protection.

## Discussion

Global climate change has enabled the expansion of mosquito populations into new ranges with concomitant increases in the variety and incidence of mosquito-borne diseases. Recent ZIKV epidemics have demonstrated the need for research focused on identifying novel and more effective targets at the vector level. Current vector control efforts involving microbiota or genetic manipulations, while promising, could be enhanced by the addition of antiviral drug strategies to ongoing control efforts.

In the present work, we evaluated the potential of IIS-targeted small molecules to reduce ZIKV infection prevalence and titers in *Ae. aegypti*. We demonstrated that the potent insulin mimetic DMAQ-B1 and the AKT inhibitor VIII synergized IIS-mediated antiviral immunity in *Ae. aegypti* to reduce ZIKV infection prevalence and titers in infected mosquitoes (**Fig 6**). While this study is not the first to identify IIS regulation of antiviral immunity, we have advanced this field by demonstrating that readily available and potent IIS-targeted small molecules induced substantial and significant antiviral immunity in *Ae. aegypti* against a clinically virulent strain of ZIKV. By targeting IIS as a mediator of two independent antiviral pathways, we reduced both infection prevalence and virus titers, outcomes predicted to reduce the likelihood of transmission. Both of these effects would be predicted to reduce ZIKV transmission by the primary vector *Ae. aegypti*. In demonstrating these effects, we have also provided a foundation for future translation of our findings to the field. We have demonstrated that exposure of DMAQ-B1 and AKT inhibitor VIII through either continuous exposure (**Fig 2**) or a single treatment (**Fig 3**) was sufficient to induce activation of both antiviral pathways independently of ZIKV infection. More importantly, even with the limited time range in which these drugs appear to induce pathway activation, there was a significant reduction in ZIKV positive- and vector competent-mosquitoes following IIS pathway activation (**Fig 4**). Specifically, we seek to advance small molecule delivery via attractive bait stations to induce IIS-mediated, broad antiviral immunity in mosquitoes that ingest these compounds.

Of particular interest for a potential field-based strategy is the broad impact that IIS appears to have across species. Previous studies have confirmed that exogenous treatment with or the endogenous effects of insulin in *D. melanogaster* and *Culex* spp. reduced replication of both WNV and DENV [14,44]. While this broad-antiviral effect is promising, further investigation is necessary to elucidate the full impact our proposed small molecules may have when introduced as a field-based strategy. Because of the limited studies on the effects of DMAQ-B1 and

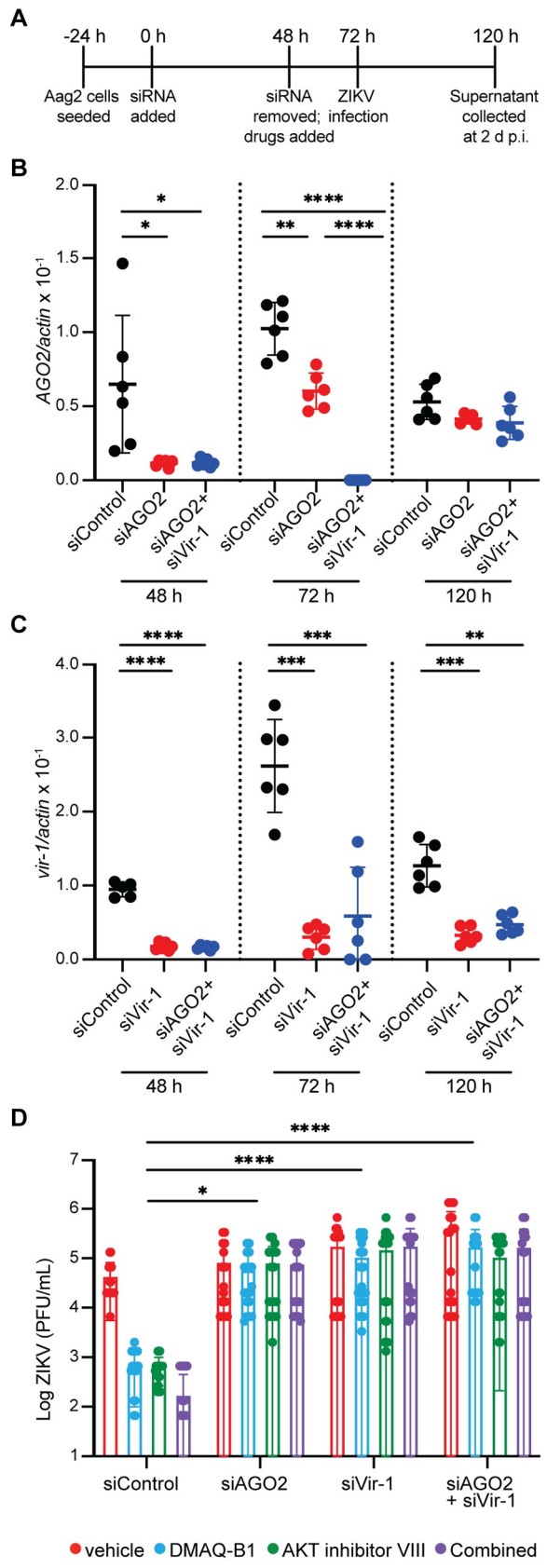

**Fig 5. Knockdown of RNAi and JAK/STAT signaling resulted in loss of drug-mediated antiviral protection.** (A) Experimental schematic illustrating process in which cells are transfected with siRNA for 48 h, drug-treated for 24 h prior to infection (72 h), and viral titer measured at 2 d p.i. (120 h). (B-C) AGO2 and vir-1 were knocked down in Aag2 cells and transcript levels were determined for (A) *AGO2* and (B) *vir-1* by qRT-PCR for cells transfected with scramble control (siControl), individual siRNA construct, or stacked siRNA (siAGO2+siVir-1) at 48 h, 72 h, and 120 h (**p<0.01; ***p<0.001; **** p<0.0001, Unpaired t test with Welch's correction). (D) 48 h following transfection, cells were primed with DMAQ-B1 or AKT inhibitor VIII for 24 h prior to infection with ZIKV (MOI = 0.01 PFU/cell). Supernatant was collected at 2 d p.i. and virus was titered by standard plaque assay (*p<0.05; **** p<0.0001,Two-way ANOVA with uncorrected Fisher's LSD test). Closed circles represent individual well replicates. Outliers were identified using a ROUT test (Q = 5%) and removed. Results are presented as pooled data of triplicate, independent experiments.

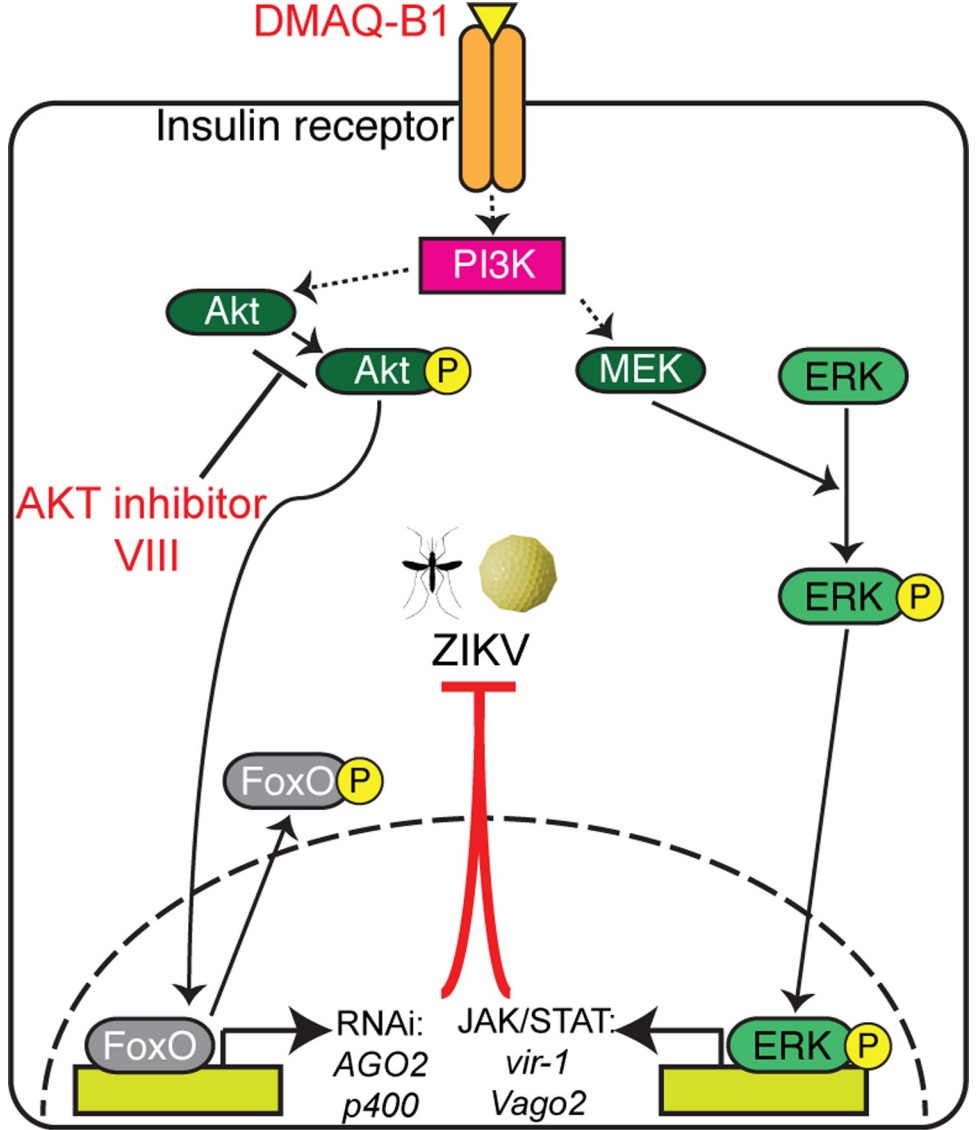

**Fig 6. Selective targeting of insulin-signaling in *Ae. aegypti* impacts canonical antiviral responses that can effectively reduce ZIKV replication and likelihood of transmission.** Schematic of proposed mechanism of antiviral action mediated by DMAQ-B1 and AKT inhibitor VIII during ZIKV infection through simultaneous induction of antiviral RNAi and JAK/STAT signaling.

AKT inhibitor VIII on insects, it is unknown if these drugs may be hazardous to beneficial species that could be exposed during mosquito treatment [13,45].

Interestingly, the antiviral effects of IIS during ZIKV infection are not limited to arthropod species. In mammalian models, ZIKV NS4A/NS4B activates PI3K-AKT signaling that is associated with neurogenetic dysregulation [46]. Further, the broadly antiviral celecoxib kinase inhibitor AR-12 and AKT inhibitor VIII have been shown to reduce ZIKV replication and pathogenesis in mice by blocking PI3K-AKT activation [47]. It is also established that diabetic individuals with dysfunctional IIS are more susceptible to severe disease during WNV [48,49], DENV [50], and ZIKV infection [51].

Given the variety of mosquito species that deploy IIS-dependent immunity against notable major arboviruses, it would be worth investigating whether similarly broad IIS regulation of antiviral responses can be detected in mammalian hosts. If so, it may be possible to develop IIS-targeted transmission blocking therapeutic drugs that mitigate Zika disease and, when delivered in blood from treated patients to *Ae. aegypti*, reduce infection in and transmission by the mosquito vector.

## Materials and methods

### Mosquito rearing

*Aedes aegypti* strain Poza Rica, from the state of Veracruz, Mexico were originally collected in 2012, and maintained as previously described [35]. Adult mosquitoes were provided continuous access to water and 10% sucrose *ad libitum* using soaked cotton balls that were replaced every other day prior to and following bloodmeal feeding. Females were allowed to feed on defibrinated sheep blood (Colorado Serum Company 31123) supplemented with 1mM ATP using an artificial feeding system to stimulate oogenesis. Larvae were reared and maintained under constant 28˚C, 70% humidity, and 12-hour light, 12 hour dark diurnal cycle. 6–9 day old adult female mosquitoes were deprived of sucrose 24 hours prior to experimental feedings as described [35]. Mosquito infections, maintenance, and plaque assays were performed under BSL3 and ACL3 facilities, approved by Colorado State University's Institutional Biosafety Committee 16-074B.

### Cells and virus

Vero cells (ATCC, CRL-81) were provided by A. Nicola and cultured at 37˚C/5% $CO_2$ in DMEM (ThermoFisher 11965) supplemented with 10% FBS (Atlas Biologicals FS-0500-A) and 1x antibiotic-antimycotic (ThermoFisher 15240062). *Ae. aegypti* Aag2 cells (*Wolbachia*-free) [41] were gifted by S. O'Neill and cultured as described in [41]. For drug treatment, culture media with 2% FBS were supplemented with 1% DMSO, 1 μM Demethylasterriquinone B1 (DMAQ-B1) (R&D Systems 1819/5), 10 μM AKT inhibitor VIII (Sigma Aldrich 124018), or combined drugs. Concentrations of DMAQ-B1 and AKT inhibitor VIII were selected at non-cytotoxic levels for both cell culture (**S1 Fig**) and adult mosquitoes (**S2 and S3 Figs**). ZIKV strain PRVABC59 (Accession # KU501215) was obtained from the CDC and was isolated in 2015 from a clinical case in Puerto Rico and prepared as described [35].

### *In vitro* virus replication

Aag2 cells were seeded into a 24-well plate at a confluency of $5 \times 10^5$ cells/well with 6 independent wells for each experimental condition. The following day, cells were treated with 1% DMSO, 1 μM DMAQ-B1, 10 μM AKT inhibitor VIII, or combined drugs in 2% FBS media as described [14] for 24 h prior to infection. Cells were then infected with ZIKV at MOI of 0.01

PFU/cell for 1 h. Virus inoculum was removed, and fresh experimental media was added. Supernatant samples were collected at 1 and 3 d p.i. for later titration. ZIKV titers were determined by standard plaque assay on Vero cells [52,53].

## Cytotoxicity of DMAQ-B1 and AKT inhibitor VIII

Cytotoxicity of DMAQ-B1 and AKT inhibitor VIII was evaluated in both cell culture and in adult female *Ae. aegypti*. DMAQ-B1 and AKT inhibitor VIII was added to a monolayer of $2.5 \times 10^5$ cells/well in 48-well plates at various concentrations (100 μM, 10 μM, 1 μM, 0.1 μM). Cells were collected at 1, 2, and 3 d post-treatment, stained with trypan blue (ThermoFisher 15250–061) and scored as live or dead as described [54]. Combined DMAQ-B1 and AKT inhibitor VIII cytotoxicity was evaluated using the maximum individual concentrations that corresponded to minimal cytotoxicity. A total of eight technical replicates were averaged for each biological replicate. 1% DMSO treated and 1% Triton X-100 treated cells were also scored as negative and positive controls, respectively. Toxicity was evaluated similarly in 6–9 day old female mosquitoes treated by either drug-treated sucrose water or drug-treated bloodmeal. Sucrose water-treated mosquitoes were provided 10% sucrose water with either 10 μM DMAQ-B1, 10 μM AKT VII inhibitor, combined treatment of both 10 μM DMAQ-B1 and 10 μM AKT inhibitor, or 1% DMSO vehicle control *ad libitum*. Mosquitoes were maintained for 14 d to monitor mortality. Bloodmeal-treated mosquitoes were fed a bloodmeal containing small molecule drugs (100 μM, 10 μM, 1 μM), 1% DMSO vehicle control, or blood only. Following 1 h of feeding, engorged females were kept and maintained on sucrose for 14 d to monitor mortality. Combined small molecule drug treatment was evaluated using observed lethal and nonlethal individual concentrations. Each experimental group contained approximately 70–100 mosquitoes.

## Immunoblotting

Protein extracts were prepared by lysing cells with RIPA buffer (25 mM Tris-HCl pH 7.6, 150 mM NaCl, 1 mM EDTA, 1% NP-40, 1% sodium deoxycholate, 0.1% SDS, 1mM $Na_3VO_4$, 1 mM NaF, 0.1 mM PMSF, 10 μM aprotinin, 5 μg/mL leupeptin, 1 μg/mL pepstatin A). Protein samples were diluted using 2x Laemmli loading buffer, mixed, and boiled for 5 minutes at 95˚C. Samples were analyzed by SDS/PAGE using a 10% acrylamide gel, followed by transfer onto PVDF membranes (Millipore IPVH00010). Membranes were blocked with 5% BSA (ThermoFisher BP9706) in Tris-buffered saline (50 mM Tris-HCl pH 7.5, 150 mM NaCl) and 0.1% Tween-20 for 1 h at room temperature.

Primary antibody labeling was completed with anti-P-Akt (1:1,000; Cell Signaling 4060), anti-P-ERK (1:1000; Sigma M8159), anti-P-FOXO (1:1000; Millipore 07–695), or anti-actin (1:10,000; Sigma A2066) overnight at 4˚C. Secondary antibody labeling was completed using anti-rabbit IgG-HRP conjugate (1:10,000; Promega W401B) or anti-mouse IgG-HRP conjugate (1:10,000; Promega W402B) by incubating membranes for 2 h at room temperature. Blots were imaged onto film using luminol enhancer (ThermoFisher 1862124). Densitometry analysis was completed using three independent blots using BioRad Image Lab and GraphPad Prism 9 with bands normalized to actin.

## RNA interference *in vitro*

Long dsRNA targeting *Ae. aegypti AGO2*, *vir-1*, and non-targeting control dsRNA was synthesized as described [41]. Targeted sequences and primers are listed in **S1 Table**. dsRNA was transfected into Aag2 cells as described [41] for 48 h prior to small molecule treatment and infection. RNA was extracted and purified to confirm reduced expression by qRT-PCR at 48

h, 72 h, and 120 h post transfection. Viral concentration was confirmed by standard plaque assay at 2 d p.i.

## Quantitative reverse transcriptase PCR

qRT-PCR was used to measure mRNA levels in *Ae. aegypti* Aag2 cells and adult females. Cells or mosquitoes were lysed with Trizol Reagent (ThermoFisher 15596). RNA was isolated by column purification (ZymoResearch R2050), DNA was removed (ThermoFisher 18068), and cDNA was prepared (BioRad 170–8891). Expression of *Ae. aegypti AGO2, p400, Vago2,* and *vir-1* were measured using SYBR Green reagents (ThermoFisher K0222) and normalized to *actin* to measure endogenous gene levels for all treatment conditions. The reaction for samples included one cycle of denaturation at 95˚C for 10 minutes, followed by 45 cycles of denaturation at 95˚C for 15 seconds and extension at 60˚C for 1 minute, using an Applied Biosystems 7500 Fast Real Time PCR System. ROX was used as an internal control. qRT-PCR primer sequences are listed in **S1 Table**.

## Immunofluorescence microscopy

*Ae. aegypti* Aag2 cells were seeded onto coverslips in 12-well plates at a confluency of approximately $1 \times 10^6$ cells/well. Cells were then treated for 24 hours with 1% DMSO, 1 μM DMAQ-B1, 10 μM AKT inhibitor VIII, or combined small molecule treatment supplemented in 2% FBS media as described [14]. Coverslips were fixed in 4% paraformaldehyde for 10 minutes at room temperature, permeabilized in 0.1% Triton-X-100 for 30 minutes at room temperature and blocked in 1% BSA in TBS for 30 min at 37˚C. Primary antibody labeling was completed with anti-P-FOXO (1:100) and anti-P-ERK (1:100) for 2 h at humified room temperature. Secondary antibody labeling was completed using anti-rabbit (Life Technologies A11034) or anti-mouse (Life Technologies A11029) Alexafluor 488 (1:300) by incubating membranes for 1 h at room temperature in the dark. Samples were stained with DAPI (1:100; Cell Signaling 4083), mounted onto coverslips using ProLong Diamond Antifade Mountant (Invitrogen P36961), and imaged using a Leica Sp8X confocal microscope. Localization percentages were determined using Adobe Illustrator 2021 by counting the total number of cells and evaluating if green-fluorescent signal was cytosolic, nuclear, or no signal in relation to DAPI-stained nuclei.

## Mosquito infections

Fresh ZIKV virus stock was made from Vero cells infected at MOI of 0.1 PFU/cell at 72 hours prior to bloodmeal feed of 6–9 day old female mosquitoes as described [35,36]. Mosquitoes were fed a bloodmeal supplemented with 1 mM ATP and infected with the fresh virus inoculum that was back-titrated to $2.4 \times 10^6$ PFU/mL. Bloodmeals included 1% DMSO vehicle control, 10 μM DMAQ-B1, 10 μM AKT inhibitor VIII, or combined drugs. Following 1 h of feeding, mosquitoes were anesthetized on ice and engorged mosquitoes were moved into new cartons and maintained on sucrose. Mosquitoes were collected at 3, 7, and 11 d p.i. in which the midgut, salivary glands, and carcass were separated, homogenized, and filtered for infection determination and viral titer. Whole mosquitoes were collected at the same timepoints for qRT-PCR analysis.

## Quantification and statistical analysis

Results presented as dot plots show data from individual biological replicates (n = 3–18), the arithmetic mean of the data, shown as a horizontal line. Biological replicates of adult mosquitoes (n = 2–24) consisted of two pooled mosquitoes. Results shown are representative of at

least duplicate independent experiments, as indicated in the figure legends. All statistical analyses of biological replicates were completed using GraphPad Prism 9 and significance was defined as $p < 0.05$. One-way ANOVA with Tukey's correction for multiple comparisons was used for densitometry analysis. Two-way ANOVA with Tukey's correction for multiple comparisons was used for microscopy localization analysis. Unpaired t test with Welch's correction was used for qRT-PCR and *in vivo* viral titer analysis. Two-way ANOVA with Tukey's correction was used for analysis of multiday *in vitro* viral titer. Two-tailed Fisher's exact test was used to compare infection prevalence. Two-way ANOVA with Tukey's correction for multiple comparisons was used for analysis of small molecule cytotoxicity *in vitro* and *in vivo*. All error bars represent standard deviation (SD) of the mean. Outliers were identified using a ROUT test (Q = 5%) and removed.

## Supporting information

**S1 Table. qRT-PCR Primers.**
(DOCX)

**S2 Table. Statistical analysis of data presented in Fig 5D.**
(XLSX)

**S1 Fig. DMAQ-B1 and AKT inhibitor VIII exhibited dose-dependent cytotoxicity in Aag2 cells.** Aag2 cells were treated with various concentrations of (A) DMAQ-B1, (B) AKT inhibitor VIII, (C) combined drugs, or DMSO vehicle control and cell viability was measured by trypan blue exclusion. Cells that received 1% Trixton-X-100 treatment were used as a positive, 100% lethality control. Closed circles represent biological replicates measured in technical triplicate. Horizontal black bars represent the mean. Error bars represent SD. Significance was measured by Two-Way ANOVA with 1% DMSO vehicle control ($^*p < 0.01$). Data are representative of triplicate independent experiments.
(TIF)

**S2 Fig. Continuous DMAQ-B1 and AKT inhibitor treatment via sucrose water does not impact mosquito survival.** Adult female *Ae. aegypti* were given sucrose water supplemented with 1% DMSO (vehicle), 10 μM DMAQ-B1, 10 μM AKT inhibitor VIII, or combined drugs *ad libitum* and toxicity was measured by survival over 14 days. Closed circles represent percent survival of mosquitoes (n = 60–100) measured in triplicate. Horizontal black bars represent the mean. Error bars represent SD. Significance was measured by Two-Way ANOVA with 1% DMSO vehicle control. Data are pooled triplicate independent experiments.
(TIF)

**S3 Fig. DMAQ-B1 and AKT inhibitor VIII exhibited minimal, dose-dependent toxicity to *Ae. aegypti* in bloodmeal.** Adult female *Ae. aegypti* were treated with various concentrations of (A) DMAQ-B1, (B) AKT inhibitor VIII, (C) combined drugs and toxicity was measured by survival over 14 days. Closed circles represent percent survival of mosquitoes (n = 60–100) measured in triplicate. Horizontal black bars represent the mean. Error bars represent SD. Significance was measured by Two-Way ANOVA with 1% DMSO vehicle control ($^*p < 0.05$). Data are representative of duplicate independent experiments.
(TIF)

**S4 Fig. RNAi and JAK/STAT signaling was induced in small molecule treated mosquitoes at 3 d p.i.** Induction of (A) *AGO2*, (B) *p400*, (C) *Vago2*, and (D) *vir-1* in adult female *Ae. aegypti* was measured by qRT-PCR 3 d p.i. of ZIKV- and drug-containing bloodmeal. ($^*p < 0.05$; $^{**}p < 0.01$; $^{***}p < 0.001$). Open circles represent individual biological replicates.

Outliers were identified using a ROUT test (Q = 5%) and removed. Horizontal black bars represent the mean. Error bars represent SDs. Data are representative of duplicate independent experiments.
(TIF)

**S5 Fig. Additional RNAi and JAK/STAT genes were induced in small molecule treated mosquitoes at 3, 7, and 11 d p.i.** Induction of additional immune genes at (A-C) 3, (D-F) 7, and (G-I) 11 d p.i. in adult female *Ae. aegypti* was measured by qRT-PCR. RNAi associated genes (A, D, G) *Dicer2* and (B, E, H) *Ppo8* and JAK/STAT (C, F, I) *dome* were measured. (*p < 0.05; **p<0.01; ***p<0.001; **** p<0.0001, unpaired t test with Welch's correction for multiple comparisons). Closed circles represent individual replicates. Outliers were identified using a ROUT test (Q = 5%) and removed. Horizontal bars represent mean and error bars represent SD. Results represent duplicate independent experiments.
(TIF)

**S6 Fig. Infection prevalence and ZIKV titers were not different among small molecule-treated and control *Ae aegypti* at 3 d p.i.** *Ae. aegypti* were primed with 1% DMSO, 10 μM DMAQ-B1, 10 μM AKT inhibitor VIII, or combined drugs and infected with ZIKV by blood-meal. Mosquitoes (n = 30) were collected at 3 d p.i. and individual midguts, pairs of salivary glands, and carcasses were prepared and titered by standard plaque assay. Infection prevalence was determined by comparing the number of mosquitoes with detectable virus to the total mosquitoes in the sample. Viral titer was measured in mosquitoes that were positive for ZIKV. There were no differences in infection prevalence or viral titers among conditions. Open circles represent biological replicates. Outliers were identified using a ROUT test (Q = 5%) and removed. Bars represent the mean. Error bars represent SDs. Data are representative of duplicate independent experiments.
(TIF)

## Acknowledgments

We thank A. Nicola, S. O'Neill, G. Ebel, and K. Olson for cells, viruses, and mosquitoes used in these experiments. We thank G. Terradas and E. McGraw for providing siRNA sequences. We also like to thank S. Bennett, C. Blair, and B. Foy, amongst others of the Center for Vector-borne Infectious Diseases, for assistance with mosquito husbandry and feedback regarding experimental design. We also thank L.R.H Ahlers for critical reading of our manuscript.

## Author Contributions

**Conceptualization:** Chasity E. Trammell, Alan G. Goodman.

**Formal analysis:** Chasity E. Trammell, Gabriela Ramirez, Irma Sanchez-Vargas, Laura A. St Clair, Oshani C. Ratnayake.

**Funding acquisition:** Chasity E. Trammell, Irma Sanchez-Vargas, Shirley Luckhart, Rushika Perera, Alan G. Goodman.

**Investigation:** Chasity E. Trammell, Gabriela Ramirez, Irma Sanchez-Vargas, Laura A. St Clair, Oshani C. Ratnayake, Rushika Perera, Alan G. Goodman.

**Methodology:** Chasity E. Trammell, Gabriela Ramirez, Irma Sanchez-Vargas, Laura A. St Clair, Oshani C. Ratnayake, Shirley Luckhart, Rushika Perera, Alan G. Goodman.

**Resources:** Chasity E. Trammell, Irma Sanchez-Vargas, Shirley Luckhart, Rushika Perera, Alan G. Goodman.

**Supervision:** Alan G. Goodman.

**Validation:** Chasity E. Trammell, Gabriela Ramirez, Irma Sanchez-Vargas, Laura A. St Clair, Oshani C. Ratnayake, Rushika Perera, Alan G. Goodman.

**Visualization:** Chasity E. Trammell, Alan G. Goodman.

**Writing – original draft:** Chasity E. Trammell.

**Writing – review & editing:** Chasity E. Trammell, Gabriela Ramirez, Irma Sanchez-Vargas, Shirley Luckhart, Rushika Perera, Alan G. Goodman.

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
