## [Decision Letter · Decision Letter 0]

31 Oct 2021

Dear Dr. Goodman,

Thank you very much for submitting your manuscript "Coupled small molecules target RNA interference and JAK/STAT signaling to reduce Zika virus infection in Aedes aegypti" for consideration at PLOS Pathogens. As with all papers reviewed by the journal, your manuscript was reviewed by members of the editorial board and by several independent reviewers. In light of the reviews (below this email), we would like to invite the resubmission of a significantly-revised version that takes into account the reviewers' comments.

As you will see, both reviewers suggest that additional experiments are necessary for PP and for fulfilling the comprehensive nature of the research as promised by the paper.

Reviewer 1 suggests really only 1 experiment. Reviewer 2 offers numerous options, suggested that several of these may suffice.

The authors will need to tackle some of these experiments for a revision to be considered.

We cannot make any decision about publication until we have seen the revised manuscript and your response to the reviewers' comments. Your revised manuscript is also likely to be sent to reviewers for further evaluation.

Sincerely,

Elizabeth A. McGraw, PhD

Associate Editor

PLOS Pathogens

Sonja Best

Section Editor

PLOS Pathogens

Kasturi Haldar

Editor-in-Chief

PLOS Pathogens

orcid.org/0000-0001-5065-158X

Michael Malim

Editor-in-Chief

PLOS Pathogens

orcid.org/0000-0002-7699-2064

As you will see, both reviewers suggest that additional experiments are necessary for PP and for fulfilling the comprehensive nature of the research as promised by the paper.

Reviewer 1 suggests really only 1 experiment. Reviewer 2 offers numerous options, suggested that several of these may suffice.

The authors will need to tackle some of these experiments for a revision to be considered.

Reviewer's Responses to Questions

**Part I - Summary**

Reviewer #1: In “Coupled small molecules target RNA interference and JAK/STAT signaling to reduce Zika virus infection in Aedes aegypti”, Trammell and colleagues utilized in vitro and in vivo models to advance the understanding of ingested vertebrate factors of the insulin/insulin-like growth factor signaling cascade could simultaneously induce both the JAK/STAT and RNAi antiviral pathways and decrease infection of Aedes aegypti by Zika virus (ZIKV). Given the dearth of vaccines and therapeutics, the only methods for controlling most arbovirus outbreaks are avoidance and abatement of mosquitoes, and use of insecticides. Therefore, there is an enormous value in methods of modulating mosquito infectability by arboviruses.

The work presented in well-conducted, utilizing appropriate controls and thoughtfully designed while the results are quite promising and are a clear advancement to the field. Despite this however, there is some concern in the design of the in vivo element of the experiment, that

Is elaborated on specifically below. In addition, minor grammatical and writing suggestions are noted.

Reviewer #2: The present paper titled ‘Coupled small molecules target RNA interference and JAK/STAT signaling to reduce Zika virus infection in Aedes aegypti’ by Trammell et al., describes the antiviral properties of two small molecules which induce the RNAi and Jak/STAT pathways. The authors use mostly qRT-PCR to detect RNAi and Jak/STAT pathway gene expression to determine the impact of drug treatment on these pathways and virus plaque assay to detect the impact on virus replication. The major conclusions that the authors make are that a combination of two drugs (which feed into insulin signaling) can increase antiviral responses, specifically RNAi and Jak/STAT, and reduce virus transmission in vivo.

The paper is generally well-written and organized. The topic of study should be of interest to readers of PLOS Pathogens, but this reviewer has some concerns with the clarity of the data and whether the conclusions are supported.

**Part II – Major Issues: Key Experiments Required for Acceptance**

Reviewer #1: 1. While the in vitro data is largely complete, there is some concern about the design of the mosquito experiment. Specifically, the two small molecules were provided to mosquitoes within the bloodmeal itself alongside the virus. Given that the authors note the real-world application of this mechanism as a field-based strategy, such an experimental paradigm is quite unrealistic. This is because mosquitoes would likely encounter the bait stations/lures impregnated with the compounds either before or after ingesting a virus laden bloodmeal. It would therefore be advantageous to present a more field accurate mosquito experiment alongside the experiment currently depicted in Figure 3.

a. Specifically, exposure to mosquitoes to the compounds via the sugar or water offered prior to exposure to the bloodmeal would strengthen the external applicability of the work.

2. The authors have framed the application of this study to mosquito control via bait stations, however in “Dong S, Dimopoulos G. Antiviral Compounds for Blocking Arboviral Transmission in Mosquitoes. Viruses. 2021 Jan;13(1):108.” it was noted that the attractive toxic sugar bait strategy “also attracted and killed nontargeted beneficial insects such as honeybees and parasitoid wasps”. A previous study “Ahlers LRH, Trammell CE, Carrell GF, Mackinnon S, Torrevillas BK, Chow CY, et al. Insulin Potentiates JAK/STAT Signaling to Broadly Inhibit Flavivirus Replication in Insect Vectors. Cell Rep. 2019 Nov;29(7):1946-1960.e5” demonstrated that the Drosophila melanogaster are also affected by insulin. While the compounds in the present study do not appear toxic at the concentrations tested, it is reasonable that these concentrations would also have consequences to other insects that sugar feed. Is there any evidence that these agents would have pleiotropic effects in non-target species causing changes to longevity, fecundity, etc? While follow-up experiments to address this are not within the purview of this paper, this limitation and gap should not be ignored in the discussion.

Reviewer #2: My major concerns are with the distinction of the two pathways (RNAi and JAK/STAT) and what the data is really supporting (see specific issues below). Some experiments indicate a much more complicated overlap between the results of the two drugs used, and Figure 4C, which would be a crucial experiment to identify the mechanism of these drugs, does not clarify but rather confuse interpretation of results. While I think the antiviral effect of the two drugs is undeniable and important, this paper is aiming at a clear mechanistic level of how these drugs work in the context of virus infection, but it is not delivering on that. As I said, my main worry comes from the questions that Figure 4C brings up (see more specific comments below). In addition, there are other functional outputs for RNAi (such as small RNA sequencing) that could have been considered to support that part of the mechanism.

I am not expecting all of the experiments mentioned below, but a set of these could strengthen this paper on the mechanistic side. I think right now the mechanism is not clear and one or two of these experiments should be considered (whichever experiments provide useful/clarifying data):

- repetition of the experiment(s) in Figure 4C

- verification of knockdown in these samples at the time when virus is measured

- quantification of other RNAi and Jak/STAT genes

- knockdown of other RNAi and JAK/STAT genes and addition to Fig 4C

- mimicking the proposed function of the drug through other experiments to indicate whether the drugs are antiviral through the proposed mechanism (so this is related to Fig 4C, but maybe starting further upstream with knockdown of FOXO, ERK, Akt, MEK, IR, etc and subsequent drug treatment)

- other readouts to test any impact on RNAi, such as small RNA sequencing (but there may be other ways to test this)

**Part III – Minor Issues: Editorial and Data Presentation Modifications**

Reviewer #1: 1. Line 38: The word “memetic” is incorrect. The correct word is “mimetic” which the authors use throughout the manuscript

2. Line 49: The statement “particularly those transmitted by mosquitoes” is unnecessary

3. Line 67: The use of the phrase “virus-free” regions is incorrect.

4. Figure 3: There appears to be mismatch between mosquito populations between prevalence and titers. In all prevalence figures (3A-C, 3G-I), 30 individual mosquitoes were screened for presence of virus. For an example, consider the vehicle condition in 3A and 3D. Figure 3A shows 33% of mosquitoes were infected with ZIKV which corresponds to 10 mosquitoes. The corresponding dot plot in D has 9 dots. This appears to be a common occurrence throughout Fig 3. Can this be clarified?

5. Lines 237-239: The phrase “…the need for host-virus interactions research” reads awkwardly. Consider rephrasing

6. Line 242: The word “therapeutic” in entirely inappropriate here

7. Line 277-278: Clarify means by which mosquitoes were given access to sugar and water

Reviewer #2: Figure 4C is highly confusing – not in its design, but the results are confusing. While results for the drug treatments look as expected in the control siRNA samples, it is highly surprising that essentially all other siRNA/drug combinations look identical in ZIKV titers. This result removes the clear explanations for how RNAi and JAK/STAT may be induced by the different drugs. For example, in Figure 1I, the authors show no significant impact of DMAQ-B1 treatment on vir-1 expression, but in Figure 4C, vir-1 silencing removed any effect that DMAQ-B1 had on ZIKV titers – so the effect was dependent on vir-1? And if the impact is so clearly vir-1 dependent, how can it be just as clearly Ago-2 dependent? Essentially, there should be more defined differences between the drug/siRNA combinations to support the model shown in Figure 4D. This figure legend also provides no information on independent replicates or even experimental replicates and this reviewer is a bit concerned about the validity of these results. Maybe some independent replicate experiments are needed to try to understand what is going on? Expression of vir-1 and Ago-2 in these exact samples at the time when virus was harvested also would have been useful (as opposed to just at 48h post transfection).

Another larger concern is the difference in this experimental design to the use of sugar-bait feeding. While I understand that this is an initial study, the authors imply that these small molecules could be sugar-fed to mosquitoes and reduce transmission – sugar feeding can change how/when/how efficiently these drugs are taken up and it would have been nice to see an experiment where mosquitoes are provided a virus containing blood-meal while being held on sugar-water (or other sugar bait) that contains the drugs – the potential constant low-level exposure with targeting to the crop instead of the midgut could impact results in either direction.

Line 193: For Figure 3D-F, the authors state that viral titers were reduced less than 2-fold, however based on the y-axis, the data is shown in a logarithmic scale and was reduced by more than 10-fold in the midguts. Carcass data was unclear/not significant, and the salivary gland data probably had too few data points. I think it might be useful to not condense these descriptions into one sentence, but clearly describe each figure panel. For 11dpi, the authors also just say that it is more than two-fold when it is really more than 10-fold (unless I am misinterpreting something).

Line 198-200: The authors compare their data to another study in which 4.8log10 PFU/mL was determined as the required titer in salivary glands to transmit ZIKV. I think the authors may need to be careful with these numbers and the subsequent sentence about ‘reducing numbers below the level of transmission’ – the reason is that all other salivary glands from control mosquitoes also had titers below this 4.8log10 PFU/mL threshold, indicating that either the methods were not as sensitive here as in the other study, or that none of these mosquitoes were transmission competent (in which case there was no reduction below the threshold since none of the mosquitoes passed it). It does not take away from these impressive results, but the wording should be adjusted a bit in my opinion.

The discussion uses and mentions many of my points above as well and would need some adjustment according to my concerns (e.g. the reduction below transmission potential in line 251, or the proposed sugar-bait feeding that was not addressed here experimentally).

Normalizing the qRT-PCR data to the control would help a bit with interpretation, especially in Figure 4A, were the knockdown of gene expression is shown. By normalizing these (ΔΔct or other fold normalization), interpretation would be more intuitive.

Figure 1H-I: The label on the x-axis says 1 day post infection, but the text and the figure legend do not indicate that these cells were infected. Either the next/legend neglected to say that ago2 and vir-1 levels were measured in infected cells, or the axis should be labeled something like ‘post-treatment’ or similar.

The pathway overview figure (Figure 4D) was very helpful – I am glad it was included, and I understand why it was the last figure of the paper, but I actually looked for something like this right away when starting to read the results, mostly in order to interpret the Western Blot results and how they matched the expectations. But I think it helped me then follow the other results (esp. Figure 1) better too. Maybe it could be introduced early on somehow? Or an adjusted version of it maybe? Just a thought because it would help understand the role of the inhibitors quickly (more so than the text). Figure 4D could be a panel in figure 1 of the paper and then the final figure could somehow link it back to the bigger organismal level - the strategy of mosquito feeding on a treated sugar bait, upregulation of RNAi and Jak/STAT, and reduced ZIKV transmission (?), assuming these mechanisms can be clarified a bit more.

Line 100: reference 40 is also relevant here as one of the first to implicate JAK/STAT in ZIKV antiviral responses – none of the cited references 17-19 seem to be from mosquito-based studies (?).

(Also, Figure 3G-I should be moved up since it these panels are mentioned first in the text.)

PLOS authors have the option to publish the peer review history of their article (what does this mean?). If published, this will include your full peer review and any attached files.

Reviewer #1: No

Reviewer #2: No
---

## [Editor Report · Decision Letter 1]

1 Mar 2022

Dear Dr. Goodman,

We are pleased to inform you that your manuscript 'Coupled small molecules target RNA interference and JAK/STAT signaling to reduce Zika virus infection in Aedes aegypti' has been provisionally accepted for publication in PLOS Pathogens.

Best regards,

Elizabeth A. McGraw, PhD

Associate Editor

PLOS Pathogens

Sonja Best

Section Editor

PLOS Pathogens

Kasturi Haldar

Editor-in-Chief

PLOS Pathogens

orcid.org/0000-0001-5065-158X

Michael Malim

Editor-in-Chief

PLOS Pathogens

orcid.org/0000-0002-7699-2064
---

## [Editor Report · Acceptance letter]

30 Mar 2022

Dear Dr. Goodman,

We are delighted to inform you that your manuscript, "Coupled small molecules target RNA interference and JAK/STAT signaling to reduce Zika virus infection in *Aedes aegypti*," has been formally accepted for publication in PLOS Pathogens.

Best regards,

Kasturi Haldar

Editor-in-Chief

PLOS Pathogens

orcid.org/0000-0001-5065-158X

Michael Malim

Editor-in-Chief

PLOS Pathogens

orcid.org/0000-0002-7699-2064